# Understanding Physiotherapy Within Older Adult Inpatient Mental Health Wards: Tailoring Physiotherapy Pathways in a Complex Environment

**DOI:** 10.3390/healthcare13243226

**Published:** 2025-12-10

**Authors:** Christopher Rushworth, Gerard Hoford, Philip Hodgson

**Affiliations:** 1Pennine Care NHS Foundation Trust, Etherow Building, Tameside General Hospital, Ashton-under-Lyne OL6 9RW, UK; 2Pennine Care NHS Foundation Trust, The Meadows, Offerton, Stockport SK2 5EQ, UK; gerard.hoford@nhs.net; 3Physiotherapy Department, Tees, Esk and Wear Valleys NHS Foundation Trust, West Park Hospital, Edward Pease Way, Darlington DL2 2TS, UK; philip.hodgson@nhs.net; 4School of Science, Technology and Health, York St John University, Lord Mayor’s Walk, York YO31 7EX, UK

**Keywords:** physiotherapy, mental health, physical health, inpatient, strength, balance, falls, ageing, lifestyle

## Abstract

**Background**: Physical health comorbidities are closely associated with mental health diagnoses, particularly among older adults. While physiotherapy services are increasingly integrated into older adult inpatient mental health settings, robust service-level evidence remains limited. **Objective**: This service evaluation aimed to assess the clinical characteristics, physical outcomes, and associative factors within a dedicated physiotherapy service in this complex setting. Secondary aims included determining the clinical utility and applicability of outcome measures for risk stratification and functional assessment, and identifying patterns related to serious injury and falls within this specific, complex older adult mental health inpatient population. **Methods**: Retrospective data was extracted from 302 patients (mean age 78.9 years) across five older adult inpatient mental health wards in one NHS trust, covering the period from January 2023 to November 2024. Data analysed included demographics, diagnoses, physical outcome measures (e.g., Hand Grip Strength, MRC-SS, and Tinetti Balance Score), falls per year, length of stay, and destination placement. **Results**: Delirium (44.4%) was the most prevalent primary diagnosis, followed by functional (40.1%) and dementia (15.6%). Physical health measures showed that patients with serious injury had lower mean Tinetti scores and a higher mean number of falls per year (*p* = 0.040 and *p* < 0.001, respectively). Intervention type was significantly associated with falls per year, with patients who received no intervention due to mental health reasons reporting more falls (*p* = 0.006). **Conclusions**: The weak association found between age and strength, balance and falls in this population emphasises the need to utilise objective assessment tools, such as the Tinetti Assessment for risk stratification, including for falls. This study emphasises the need for tailored service pathways and active patient involvement towards discharge planning, and the essential role of physiotherapy integration within the multidisciplinary team. Future research remains essential in demonstrating the effectiveness of physiotherapy intervention and building the evidence base for physiotherapy in older adults’ mental health inpatient settings.

## 1. Introduction

The escalating global prevalence of mental health conditions presents a profound public health challenge [1], with a particularly significant impact on the older adult population [2]. Among these, Serious Mental Illness (SMI), encompassing conditions such as severe depression, bipolar disorder, and schizophrenia, carries a disproportionately high burden, often leading to profound functional impairment and reduced quality of life [3]. As societies worldwide experience a demographic shift towards an increasingly aged population [4], the intricate and often bidirectional connection between physical and mental well-being in this vulnerable group is likely to become an area of increasing focus [5].

Older adults frequently contend with complex comorbidities [6], polypharmacy [7], frailty [8], and cognitive impairments [9], all of which can exacerbate mental health issues and compromise their functional independence [10]. Compounding these challenges are the synergistic effects of SMI, physical illness, and psychotropic medications, which frequently cause impaired mobility, and increased fall risk in this specific population [11,12]. Individuals with SMI often experience significantly higher rates of physical health issues [13], including cardiovascular disease [14], diabetes [15], and respiratory conditions [16], leading to a reduced life expectancy compared to the general population [17,18]. This disparity is frequently attributed to a combination of factors, including to the side effects of psychotropic medications [19], which can contribute to metabolic syndrome and weight gain [20], as well as lifestyle factors such as poor diet [21], sedentary behaviour [22], and reduced access to physical healthcare services [23]. It is crucial to acknowledge the wide variety of interconnected variables affecting inpatient presentations, including the severity and chronicity of mental health conditions [24], concurrent physical health issues [25], medication regimens [26], and even environmental factors such as staffing levels and ward environments [27], all of which influence a patient’s physical well-being and response to interventions [28].

Within this complex patient setting, physiotherapy, traditionally known for its contributions to physical rehabilitation, is becoming increasingly acknowledged for its integral and evolving role in providing holistic mental healthcare [29]. This encompasses a broad spectrum of interventions aimed at improving functional mobility [30], promoting engagement in physical activity, enhancing balance and strength [31], and promoting overall well-being [32]. These elements are of high importance for older individuals navigating the complexities of mental illness, as physical activity and functional independence are strongly linked to improved mood [33,34], cognitive function [35], and reduced symptoms of mental health disorders [36]. Physiotherapy professionals, integrated within multidisciplinary mental health teams, are uniquely positioned to address an individual’s physical symptoms and associated mental health decline, advocating for an active and engaged recovery [37].

Despite this growing recognition within clinical practice, the existing body of literature reveals a lack of robust evaluations of specific physiotherapy service models operating within older people’s mental health inpatient settings [38]. While the general principles and benefits are understood, the efficacy and specific impact of dedicated physiotherapy services delivered in real-world psychiatric inpatient environments for older adults remain underexplored [39].

Research often focuses on specific interventions or outcomes in broader populations, neglecting comprehensive service evaluations that can inform best practice in specialised settings [40]. This gap in knowledge is particularly important given the unique challenges presented by an inpatient mental health environment, including the diverse diagnoses [41], fluctuating cognitive states [42], and varied physical capabilities of older patients [43].

Understanding how a physiotherapy service operates within such a context, and what its tangible benefits are, is crucial for optimising care pathways and ensuring resource allocation is evidence-based [44]. While fall risk is well documented in acute elderly care, there is a significant gap in understanding the clinical utility of physical assessment tools (like the Tinetti Assessment and Hand Grip Strength) in older people’s mental health wards, which necessitates this service evaluation. There is a clear need for studies that bridge the gap between theoretical understandings of physiotherapy’s benefits and its practical application and measurable impact within established older adult mental health services [45].

This study sought to address this critical gap by conducting a comprehensive evaluation of the characteristics and outcomes associated with a physiotherapy service delivered across older people’s inpatient mental health wards within the Pennine Care NHS Foundation Trust. The overarching aim was to assess the broader impact of this integrated physiotherapy service. To achieve this, the service evaluation specifically aimed as follows: (1) to describe the clinical and demographic characteristics of the older adult mental health inpatient cohort receiving physiotherapy; (2) to assess the clinical utility and applicability of objective physical assessment measures (Hand Grip Strength, Medical Research Council Sum-Score, and the Tinetti Assessment) for risk stratification, particularly for falls and serious injury; and (3) to identify the factors associated with physiotherapy engagement and successful discharge outcomes, and use these findings to inform the development of tailored physiotherapy pathways within this complex environment.

The findings from this comprehensive service evaluation provide invaluable insights that extend beyond the immediate context of the Pennine Care NHS Foundation Trust. The results provide valuable data and transferable insights to inform the development of evidence-based clinical pathways and practice guidelines for similar older adult mental health inpatient settings. By systematically exploring the intricate correlations between demographic factors, baseline physical health indicators, and key patient outcomes, this study attempts to inform and enhance physiotherapy service delivery. The results are expected to contribute to the development of more targeted and effective interventions, optimised resource allocation, and the strengthening of the evidence base for the integral role of physiotherapy within older adult mental health services.

## 2. Materials and Methods

### 2.1. Study Setting and Design

This study is a retrospective exploratory cohort analysis of routine service evaluation data. The project was registered and approved as a Service Evaluation by the Research and Innovation Team at Pennine Care NHS Foundation Trust. All data used in the study was retrospectively extracted from clinical records by the lead author who is a specialist physiotherapist working within older adult mental health inpatient wards. Because the data was obtained from routine assessments with no changes to normal practice, the Trust deemed that informed consent was not applicable.

Data analysis was completed using data from the older adult inpatient mental health wards within the Pennine Care NHS Foundation Trust. We collected a range of data to provide a comprehensive overview of an individual’s presentation. This included information such as age, diagnoses, balance, Hand Grip Strength, muscle strength, falls per year, and length of stay. Data was recorded in a Microsoft Excel database. The inclusion criteria were as follows. (1) Ward Type: Patients admitted to five older adult mental health wards within the study period who received a formal initial physiotherapy assessment, typically within 72 h of referral to physiotherapy. (2) Minimum Age: Patients were generally over 65 years old, aligning with the service specification for older adult mental health wards (the exceptions being a small number of dementia patients who were under 65 but admitted to a dementia ward). (3) Repeat Admissions: Where a patient had multiple admissions within the study period, only the first complete admission record was included in the final analysis (*N* = 302) to ensure that patients were only included once in the dataset.

The exclusion criteria were as follows: patients who were only seen for a one-off consultation (e.g., equipment review) without the need for a full assessment.

Data was collected from every patient that was referred for a physiotherapy assessment across five acute older adult mental health wards. The data collection period was from January 2023 to November 2024. The five older adult wards consisted of two dementia wards (one male and one female), two functional wards (one male and one female), and one delirium ward (mixed sex). Patients are admitted to the acute mental health wards for an assessment of their mental health condition and needs. The delirium ward admits patients with a delirium diagnosis who are unsafe to be in the community due to their current presentation. Details of ward type, bed numbers, and physiotherapy provision are shown in Table 1.

The delirium ward has a shorter average length of stay, a higher number of admissions, and therefore a higher turnover than the other wards. Due to the prevalence of high rates of physical health comorbidities, acute physical declines, and associated mobility impairments in patients with delirium, all new admissions to the delirium ward are automatically assessed by physiotherapy services (proactive pathway), usually within 48–72 h of admission. By comparison, the four mental health wards use a reactive pathway, where physiotherapy services only complete an initial assessment following a specific referral trigger. Indicators for referral to physiotherapy on these wards include impaired mobility, falls history, pain, or physical health concerns. Due to these factors, we receive more referrals from our delirium ward relative to each of the other four wards, as shown in Table 2.

We collected data on many variables and characteristics to explore how each of these affect patient presentations and outcomes. Data collected included the following: ward admitted to, primary and secondary mental health diagnosis, mobility, mean Hand Grip Strength, Medical Research Council Sum-Score [46,47], Tinetti Balance Score [48], whether the patient had intervention from physiotherapy, number of falls per year, serious injury from a fall, total length of stay, and destination placement.

### 2.2. Data Variables and Outcome Measures

Hand Grip Strength was measured using handheld Saehan Squeeze Dynamometers in the standard testing position (seated with the elbow at 90 degrees) and repeated three times on each hand. The Medical Research Council Sum-Score is derived from the bilateral assessment of six muscle groups (shoulder abductors, elbow flexors, wrist extensors, hip flexors, knee extensors, and ankle dorsiflexors) using the Oxford Scale. We used the original Tinetti Performance-Oriented Mobility Assessment which is measured using a 28-point scale (16 points for balance and 12 points for gait). The clinical cut-off scores used by the service for the Tinetti Assessment are as follows: <19 indicates a high risk of falls, 19–24 indicates a moderate risk of falls, and >24 indicates a low risk of falls.

To make each patient’s number of falls comparable, the number of falls per year was calculated as though each patient had stayed on the ward for one year. This was calculated by dividing 365 by the total length of stay, then multiplying it by the number of falls the patient had during their inpatient stay.

Serious injury was defined as any fall-related injury requiring transfer to the acute general hospital for further specialist management, specifically including fracture, intracranial haemorrhage, or sub-dural haematoma, as determined by official documentation.

The interventions were classified as yes, the patient had intervention from physiotherapy, no, the patient was already at baseline function, or no, the patient was unable to comply with intervention due to their mental health. Physiotherapy intervention is normally on a weekly basis for as long as each patient remains on the physiotherapy caseload. Patients are discharged from the caseload when they have reached their baseline level of function, are no longer receiving any benefit from physiotherapy intervention or consistently do not engage with physiotherapy intervention. Intervention can take various forms including but not limited to assessment and treatment of musculoskeletal conditions, falls assessments and treatments, complex moving and handling assessments, strength and balance programmes, mobility re-education, and transfer practice.

### 2.3. Data Analysis

Data was anonymised at the point of data analysis. Our physiotherapy team developed a list of research questions by identifying likely correlations between variables. The Excel database, along with our research questions, were sent to a data analyst who provided the initial data analysis. Following a number of meetings, the tests and variables compared were refined to provide the analysed data.

Due to the complex nature of the older adult mental health inpatient setting, there were multiple patients who were unable to engage with specific outcome measures. Specific patient data sets were not included in any test where one of the variables being analysed had missing data. In total, 23 patient datasets (out of 302) were missing at least one outcome measure score (Hand Grip Strength, Medical Research Council Sum-Score, or the Tinetti Performance-Oriented Mobility Assessment) because the patient was unable to fully engage in the initial physiotherapy assessment. Of these 23 patients, 14 had a primary diagnosis of dementia, 7 of delirium and 2 of functional (the full dataset can be made available on request).

All data has been analysed using SPSS version 29. Tests used in the data analysis include one-way ANOVAs, the *t*-test, and the chi-squared test. The initial assessment of continuous variables (Age, HGS, and Tinetti) via the Kolmogorov–Smirnov test revealed a predominantly non-normal distribution of the data. Consequently, the primary analyses employed non-parametric equivalents: the Mann–Whitney U test for two-group comparisons, the Kruskal–Wallis H test for multiple-group comparisons, and Spearman’s Rho for correlation analysis, ensuring methodological consistency with the data’s distribution. In the few instances where parametric tests were applied, Levene’s test was used to confirm homogeneity of variances.

Where ANOVA indicated a significant effect, post hoc comparisons were conducted using Tukey’s Honestly Significant Difference (HSD) test to identify which groups differed. Effect sizes for ANOVA were calculated using eta-squared (η^2^), and Cohen’s d was used for *t*-tests.

## 3. Results

### 3.1. Demographics and Clinical Characteristics

Data was collected from 302 patients (48% male, 52% female; mean age = 78.9 years, range = 59–98) across two hospital sites and five wards between January 2023 and November 2024. All 302 patients included in this service evaluation received an initial physiotherapy assessment and met the inclusion criteria. The primary diagnoses for the 302 patients were recorded as either ‘functional’, ‘delirium’, or ‘dementia’, based on the wards these patients were placed, with delirium (44.4%) being the most frequent, followed by functional (40.1%) and dementia (15.6%). Demographic information is detailed in Table 3.

A wide range of secondary mental and physical diagnoses was documented (see Appendix A), with 81 (26.8%) patients having no formal secondary diagnosis and the remainder presenting with varied neuropsychiatric or cognitive comorbidities (e.g., Alzheimer’s, Parkinson’s, and psychosis). Table 4 presents a condensed view of secondary physical health diagnoses, grouped into clinically relevant categories for efficiency and discussion. Secondary diagnoses were grouped by the authors to align with their impact on a patient’s physical and mental health. Please see Appendix A for the comprehensive list of all diagnoses observed in the cohort.

### 3.2. Baseline Measures and Correlations

Table 5 details the Pearson correlations between all core variables: age, mean Hand Grip Strength (HGS), Medical Research Council Sum-Score (MRC-SS), Tinetti Balance score, inpatient falls, falls per year, and length of stay.

Age demonstrated a weak negative correlation with mean HGS (r = −0.206, *p* < 0.001), MRC-SS (r = −0.196, *p* < 0.001), and the Tinetti scores (r = −0.381, *p* < 0.001). Additionally, age showed a weak positive correlation with falls per year (r = 0.158, *p* = 0.006) and a weak negative correlation with length of stay (r = −0.132, *p* = 0.022).

Mean HGS was moderately positively correlated with MRC-SS (r = 0.441, *p* < 0.001) and the Tinetti scores (r = 0.425, *p* < 0.001). The Tinetti scores also demonstrated a moderate positive correlation with MRC-SS (r = 0.544, *p* < 0.001) and a weak negative correlation with falls per year (r = −0.190, *p* < 0.001).

Inpatient falls were moderately to strongly associated with falls per year (r = 0.646, *p* < 0.001) and weakly to moderately associated with length of stay (r = 0.387, *p* < 0.001). Furthermore, MRC-SS showed a weak positive correlation with length of stay (r = 0.139, *p* = 0.021).

### 3.3. Serious Injury and Clinical Variables/Functional Measures

Independent samples *t*-tests examined differences in physical performance and fall risk between patients who experienced a serious injury (a fracture or brain haemorrhage as a result of a fall) during admission (*n* = 11) and those who did not (*n* = 291). Patients without a serious injury had a higher mean Tinetti score than those with a serious injury, t(280) = 1.752, *p* = 0.040. The mean number of falls per year was lower in patients without a serious injury compared to those with a serious injury, t(300) = −3.47, *p* < 0.001.

There was no significant difference in MRC-SS scores between patients with and without a serious injury, t(272) = −0.229, *p* = 0.409. Mean Hand Grip Strength did not differ significantly between groups, t(300) = 0.490, *p* = 0.312.

Patients without a serious injury were younger than those with a serious injury, t(300) = −1.81, *p* = 0.036. For falls/year × sex, Levene’s test indicated unequal variances (F = 7.13, *p* = 0.008), so Welch’s *t*-test was used; males had a higher mean number of falls per year than females, t(261.05) = 2.41, *p* = 0.008. All other tests assumed equal variances based on non-significant Levene’s tests. Table 6 demonstrates these results.

A significant effect of diagnosis was found on the mean number of falls per year, F(2, 297) = 13.56, *p* < 0.001, η^2^ = 0.084. Post hoc comparisons using Tukey HSD indicated that patients with dementia had significantly more falls per year than those with a functional diagnosis (mean difference = 9.5, SD ≈ 1.82, *p* < 0.001) and those with delirium (mean difference = 7.63, SD ≈ 1.80, *p* < 0.001). The proactive referral model on our delirium ward, whereby all patients receive a physiotherapy assessment on admission, likely contributes to a greater mean difference in falls per year between patients with dementia and those with delirium, as all delirium ward patients are assessed, not just those already identified as needing a physiotherapy assessment. If the delirium ward had used an active referral model like the four other wards, there would likely have been a higher mean falls per year on the delirium ward. Destination placement was significantly associated with several measures. The Tinetti scores varied between placement groups, F(5, 213) = 7.95, *p* < 0.001, with higher scores observed in home placements, in 24 h care, and in mental health ward groups compared to in acute hospital placement. The summary means for the Tinetti scores by destination placement are as follows: home (M = 7.37, SD = 1.33), 24 h care (M = 6.14, SD = 1.95), mental health ward (M = 9.21, SD = 2.40), and those who died (M = 9.71, SD = 3.33), with acute hospital serving as the reference group. MRC-SS scores also differed significantly by destination, F(5, 207) = 2.92, *p* = 0.014. Post hoc analysis indicated that patients discharged home had significantly higher MRC-SS scores compared to those discharged to an acute hospital (mean difference = 3.51, SD = 1.08, *p* = 0.017). Falls per year varied significantly between placement groups, F(5, 224) = 6.76, *p* < 0.001, with fewer falls at home, in 24 h care, and in mental health ward placements compared to those who had died. In contrast, mean Hand Grip Strength for both hands did not significantly differ between destination groups, F(5, 224) = 2.11, *p* = 0.06.

Intervention type was significantly associated with the mean number of falls per year, F(2, 298) = 5.16, *p* = 0.006. Those who received no intervention due to mental health reasons reported more falls than those who received an intervention or were already at baseline. The mean length of stay did not significantly differ by intervention type, F(2, 298) = 1.19, *p* = 0.306.

Finally, the ward which patients were admitted was significantly associated with the mean number of falls per year, F(5, 294) = 5.49, *p* < 0.001. Patients on the male dementia ward experienced more falls than those on the male functional, female functional, and delirium wards. Table 7 details the relationships between these variables and the outcomes/factors.

Table 8 shows the results from the chi-squared tests with the variables compared. A statistically significant association between intervention and destination placement was found, χ^2^(10) = 20.7, *p* = 0.023.

No statistically significant association was found between serious injury and diagnosis, χ^2^(2) = 0.77, *p* = 0.680. The relationship between falls per year and sex was also not significant, χ^2^(124) = 119.35, *p* = 0.601. Therefore, while males demonstrated higher mean falls per year when using the *t*-test, this association was not consistently observed across analytic approaches.

## 4. Discussion

This study sought to address the critical gap in the literature surrounding the application of a physiotherapy service in the older adult mental health inpatient setting by conducting a retrospective exploratory cohort analysis of routine service evaluation data. The overarching aim was to assess the broader impact of this integrated physiotherapy service. To achieve this, we aimed to analyse demographics and clinical profiles of older adult inpatients, examine the applicability of physical health outcome measures such as HGS, MRC-SS, and Tinetti, and investigate the associations between physiotherapy interventions and patient outcomes, including fall frequency, length of stay, and destination upon discharge. Studies have previously demonstrated the lack of robust evaluations of physiotherapy service models in this setting, whilst the specific impact of physiotherapy interventions in psychiatric inpatient environments for older adults remains underexplored [36,37].

Although the principles of physiotherapy are widely known and well researched in other populations, there is little evidence specifically around their efficacy in mental health. The effect that confounding factors have on the physical health outcomes in this population are considered in detail later in this discussion. However, it is not widely known to what extent a patient’s mental health presentation influences key physiotherapy outcome measures, including muscle strength, balance, and falls. The results in this study will be interpreted in the context of the existing literature and clinical practice. Advantages of this study are as follows: (1) its relatively large sample size for this specialist setting (*N* = 302); (2) its nature as one of the first comprehensive service evaluations of physiotherapy in older adult mental health inpatient care, providing highly relevant and generalisable operational data; and (3) its implications for practice in this specialist setting moving forward.

### 4.1. Relationship Between Age, Physical Outcome Measures, and Mental Health

Our study found that age demonstrated a weak negative correlation with Hand Grip Strength, muscle strength (MRC-SS), and balance (Tinetti Assessment) scores. Additionally, age showed a weak positive correlation with falls per year. Research connecting age with strength, balance, and falls in the older population is widely available [49]. However, the weak correlations found in our study are likely to be indicative of the increased prevalence of non-age-related factors that affect people’s physical health in the older people’s mental health setting [50]. Mental health diagnosis, psychotropic medications, sedentary lifestyle, poor diet, and frailty, alongside a higher prevalence of physical comorbidities, may all have a significant impact on a patient’s physical health. The weak correlations support the idea that biological age (driven by frailty, physical comorbidity, and lifestyle) is a more significant determinant of function than chronological age in a complex cohort like that of older adult mental health inpatients [51]. Chronological age is shown to be a poor predictor of strength, balance, and falls in this population, which emphasises the need to use objective assessment tools (HGS, MRC-SS, and Tinetti) for risk stratification, instead of relying solely on age.

### 4.2. Tinetti Assessment, Fall Risk, and Serious Injury

Patients who sustained a serious injury from a fall whilst an inpatient were found to have lower scores on the Tinetti Performance-Oriented Mobility Assessment, and a higher number of falls per year. This mirrors other studies which looked at participants over 65 without a mental health diagnosis and studies validating the link between Tinetti scores and rate of falls [48,52]. The Tinetti Assessment should therefore be used by healthcare professionals to provide an early and accurate indication of patients within this setting who will have an increased risk of falls and subsequent serious injury. Strategies can then be put in place to mitigate individual fall risk factors and reduce the likelihood of serious injury.

### 4.3. Physiotherapy Intervention and Falls Risk

We found intervention type to have a significant correlation with falls per year, with those patients who received no intervention due to mental health reasons reporting more falls than those who received intervention or those already at their baseline. There is a high likelihood that patients who received no intervention due to their mental health are a separate subgroup that have increased levels of confusion and agitation, and increased rates of delirium, all of which would increase their fall risk. This would support the need for comprehensive, multidisciplinary, multifactorial fall risk assessments [53] to mitigate individual fall risk factors, and the tailoring of physiotherapy services in this setting to increase their presence within ward rounds and safety huddles to ensure comprehensive, multidisciplinary team discussions regarding individual fall risk mitigation strategies.

Inpatient falls were weakly to moderately associated with length of stay, whilst age, HGS, MRC-SS, and Tinetti all demonstrated weak or no correlation with length of stay. Neither the MRC-SS nor Hand Grip Strength showed any association with serious injury. Confounding factors for length of stay are likely to include mental health presentation, social care provision, and bed availability in the community. Inpatient falls, HGS, MRC-SS, and Tinetti scores are all heavily influenced by non-age-related factors. Psychotropic medications, such as benzodiazepines and antipsychotics, are frequently used to treat mental health conditions including anxiety, depression, and psychosis in this population. However, falls are often attributed to common side effects like orthostatic hypotension, sedation, and extrapyramidal symptoms [54]. We strongly recommend that all fall-related interventions include a thorough review of a patient’s psychotropic medications, with particular attention to new medications and dose changes [55].

Delirium, which has been widely linked with increased falls rates [56], is known to have a higher prevalence in over-65-year-olds (1–2%), increasing to a 10% prevalence in over-85-year-olds, with a significant increase in those with a dementia diagnosis (up to 22%) [57]. The delirium ward patients had lower mean number of falls per year than our dementia ward patients, which is likely because of our proactive referral pathway on the delirium ward, where it is mandatory for all patients to be assessed on admission, compared to the reactive referral pathway on our dementia wards, where patients are referred to physiotherapy when indicated. Widely available research and NICE Guidelines suggest that falls and delirium are inextricably linked [58,59], and it is for these reasons that we recommend the implementation of two distinct physiotherapy referral pathways: (1) a mandatory, proactive pathway (like the delirium ward model) for high-risk patients (e.g., those with acute confusion or severe physical comorbidity) and (2) a targeted, reactive pathway for others based on specific functional decline or fall history.

### 4.4. Discharge Destination, Function, and Engagement

Destination placement showed a significant association with various measures including the Tinetti Assessment, with lower scores observed in those going to an acute hospital placement, compared with patients going to their home, 24 h care, or another mental health ward. Associations between MRC-SS and falls per year also differed significantly between destinations. Patients who were discharged to their home, to 24 h care, or to other mental health wards had fewer falls compared to those who had died. This would indicate that patients who have impaired balance, less strength, and more falls have a higher level of frailty, which appears to have a significant effect on their discharge destination and in some cases likely inhibits a successful discharge back to their home or to 24 h care.

The significant association between intervention and destination placement would indicate that those patients who engage in physiotherapy or are already at their baseline presentation have a greater chance of being discharged home or to 24 h care. Engagement in physiotherapy assessment is likely to be a proxy for clinical response, compliance, and overall stability (e.g., reduced delirium/agitation), which collectively enable a safer and quicker discharge. The correlation found between intervention and destination placement might be an association, rather than a direct causal effect. The significant association should be interpreted as an indicator for the importance of active patient involvement towards discharge planning, which includes tailoring physiotherapy intervention to focus on the patient’s goals for discharge [60].

### 4.5. Limitations

Although efforts were made throughout the study to minimise limitations, the study was designed as a retrospective exploratory cohort analysis of routine service evaluation data, which cannot establish direct cause-and-effect relationships or infer causality due to the non-randomised nature. Reliance on clinical documentation, non-standardised referral triggers (except the delirium ward) and missing data are all factors associated with this study design which would limit its generalisability. There is a lack of control for confounding factors, which is typical of retrospective analyses. These include variables like the severity and duration of mental illness, polypharmacy/psychotropic drug load, and the variability of clinical decisions regarding patient engagement and discharge planning, which may influence observed outcomes (e.g., successful discharge). Due to the use of heterogeneous referral pathways (proactive and reactive) the four wards using the reactive referral pathway likely excluded patients with no mobility issues by not referring them to physiotherapy, potentially skewing the dementia and functional scores observed. The difference in referral method between the delirium ward (all patients assessed following admission) and the four mental health wards (active referral method) may limit the data’s generalisability. Since patients were assessed within five distinct inpatient environments, the assumption of the independence of observations may be partially violated, as environmental factors or local clinical cultures specific to each ward could influence patient characteristics and outcomes.

Patients may have had their diagnosis altered during their admission, which could affect the data’s reliability. Some participants had missing data for some outcome measures (Hand Grip Strength, MRC-SS, and the Tinetti Assessment). In most cases this was due to the patient not being able to comply with the assessment due to their mental health. The highly complex and skewed nature of clinical data in this population makes achieving perfect statistical assumption fulfilment difficult. There was added complexity when interpreting the data for patients who died, as the dataset lacks context for the cause of death. It should be noted that patients who did not receive intervention due to mental health reasons may also represent a subgroup with greater illness severity and frailty; therefore; there is a high likelihood of confounding factors contributing to higher fall rates within this subgroup.

### 4.6. Future Research

A key finding from this research is the weak association between a patient’s age and their strength and balance, likely due to a myriad of contributory factors in the inpatient mental health setting. Future research should aim to expand our knowledge of these contributory factors, for example, looking at the effect psychotropic medications have on patients’ posture and mobility.

There remains minimal evidence associating the impact of physiotherapy interventions with patient outcomes in older adult mental health inpatient settings. Future studies could look at the association between patient outcomes and physiotherapy intervention by comparing the number of physiotherapy patient contacts with health outcomes, such as falls per year and discharge destination. Further investigation is required into the reasons for physiotherapy referrals in this setting, which will then help to shape inpatient pathways.

To optimise physiotherapy services within the mental health setting, we suggest that future research gathers patient and staff perspectives on the physiotherapy service and how it is integrated with the multidisciplinary team. Additionally, we recommend a study that compares a unit with dedicated physiotherapy to one without, to more clearly demonstrate the service’s efficacy.

### 4.7. Recommendations for Service Delivery and Policy

Hospital management and policy makers should consider this study’s findings when allocating resources and staffing for older adult mental health inpatient settings. We recommend the implementation of two distinct physiotherapy referral pathways: (1) a mandatory, proactive pathway (like the delirium ward model) for high-risk patients (e.g., those with acute confusion or severe physical comorbidity), and (2) a targeted, reactive pathway for others, based on specific functional decline or fall history. We also propose the development of formal training for multidisciplinary teams to enhance their understanding of the indications for referral to physiotherapy when using the reactive referral pathway, and the role of physiotherapy in improving physical and mental well-being.

Age is shown to be a poor predictor of strength, balance, and falls in this population, which emphasises the need to use objective assessment tools (HGS, MRC-SS, and the Tinetti Assessment) for risk stratification of physical health and falls, instead of relying solely on age. This would also support the need for (1) comprehensive, multidisciplinary, multifactorial fall risk assessments, and (2) the tailoring of physiotherapy services in older adult mental health inpatient settings to increase their presence within ward rounds and safety huddles, to ensure comprehensive multidisciplinary team discussions regarding individual fall risk mitigation strategies.

We recommend the development and formalisation of standardised, evidence-based physiotherapy pathways and guidelines for this complex setting, which should mandate the routine incorporation of objective measures, like the Tinetti Assessment, for early risk identification.

## 5. Conclusions

The weak association found between age and strength, balance and falls in this population emphasises the need to utilise objective assessment tools (HGS, MRC-SS, and Tinetti) for risk stratification, including for falls. There was a significant association between patients’ engagement in physiotherapy and their greater chance of successful discharge, which is an indicator of the importance of active patient involvement towards discharge planning. Finally, this study emphasises the need for tailored physiotherapy pathways in this complex setting and the integration of physiotherapy within the multidisciplinary team. Future research remains essential in demonstrating the effectiveness of physiotherapy intervention and building the evidence base for physiotherapy in older adult mental health inpatient settings.

## Figures and Tables

**Table 1 healthcare-13-03226-t001:** Breakdown of the wards by patient group and bed numbers.

Type	Gender	Number of Beds	Whole Time Equivalent Physiotherapy per Week
Dementia	Female	10	0.50
Dementia	Male	11	0.25
Functional	Female	20	0.50
Functional	Male	16	0.25
Delirium	Mixed	23	0.50

**Table 2 healthcare-13-03226-t002:** Admissions, length of stay, and number of physiotherapy referrals per ward during the study period.

Ward	Total Number of Admissions	Average Length of Stay for All Patients	Average Length of Stay for Referred Patients	Number of New Patients Referred to Physiotherapy
Dementia Female	61	144.9	139.6	24
Dementia Male	50	110.6	131.9	27
Functional Female	133	98.8	121.0	62
Functional Male	99	87.4	93.6	57
Delirium	167	71.6	78.7	132

**Table 3 healthcare-13-03226-t003:** Demographics and primary diagnosis.

Characteristic	Value
Sample Size	302 patients
Age (mean ± SD)	78.9 ± 8.0 years (range: 59–98)
Sex Distribution	145 Male (48.0%) and 157 Female (52.0%)
Primary Diagnosis	Delirium 134 (44.4%), Functional 121 (40.1%), and Dementia 47 (15.6%)

**Table 4 healthcare-13-03226-t004:** Secondary diagnoses and their count.

Secondary Diagnosis	Number
Anxiety and/or depression	81
Awaiting diagnosis, including probable dementia	9
Bipolar affective disorder	17
Cognitive impairment	4
Dementia (Alzheimer’s, vascular, and Lewy-body)	60
Learning disability	2
Mild cognitive impairment	3
Mixed dementia	15
Paranoid disorder	4
Parkinson’s	3
Psychosis	8
Psychotic depression	4
Schizophrenia/Schizoaffective disorder	14
Other	9

**Table 5 healthcare-13-03226-t005:** Pearson correlations and significance for variables measured.

Variable 1	Variable 2	r-Value	*p*-Value
Age	Mean HGS (Both Hands)	−0.206	<0.001 *
Age	MRC-SS	−0.196	<0.001 *
Age	Tinetti Score	−0.381	<0.001 *
Age	Falls Inpatient	0.014	0.815
Age	Falls per Year	0.158	0.006 *
Age	Length of Stay	−0.132	0.022
Mean HGS	MRC-SS	0.441	<0.001 *
Mean HGS	Tinetti Score	0.445	<0.001 *
Mean HGS	Falls Inpatient	−0.070	0.225
Mean HGS	Falls per Year	−0.094	0.104
Mean HGS	Length of Stay	−0.004	0.950
MRC-SS	Tinetti Score	0.544	<0.001 *
MRC-SS	Falls Inpatient	0.107	0.078
MRC-SS	Falls per Year	−0.010	0.871
MRC-SS	Length of Stay	0.139	0.021
Tinetti Score	Falls Inpatient	−0.114	0.057
Tinetti Score	Falls per Year	−0.190	<0.001 *
Tinetti Score	Length of Stay	0.098	0.098
Falls Inpatient	Falls per Year	0.646	<0.001 *
Falls Inpatient	Length of Stay	0.387	<0.001 *
Falls per Year	Length of Stay	0.000	0.998

Note: * indicates where two variables are statistically significant.

**Table 6 healthcare-13-03226-t006:** Independent *t*-test results for clinical variables by group.

Variable	Groups Compared	Mean ± SD (Group 1)	Mean ± SD (Group 2)	t(df)	*p*-Value
Tinetti score × Serious Injury	No injury (*n* = 272) vs. Serious injury (*n* = int)	18.13 ± 7.90	13.70 ± 6.40	1.752 (280)	0.040 *
Falls/Year × Serious Injury	No injury (*n* = 291) vs. Serious injury (*n* = 11)	5.54 ± 10.76	17.05 ± 12.23	−3.47 (300)	<0.001 *
MRC-SS × Serious Injury	No injury (*n* = 264) vs. Serious injury (*n* = 10)	54.27 ± 5.80	54.70 ± 5.25	−0.229 (272)	0.409
HGS × Serious Injury	No injury (*n* = 291) vs. Serious injury (*n* = 11)	7.26 ± 5.15	6.48 ± 4.70	0.490 (300)	0.312
Age × Serious Injury	No injury (*n* = 291) vs. Serious injury (*n* = 11)	78.76 ± 8.01	83.18 ± 7.15	−1.81 (300)	0.036 *
Falls/Year × Sex	Male (*n* = 145) vs. Female (*n* = 157)	7.55 ± 12.57	4.49 ± 9.12	2.41 (261.05)	0.008 *

Note: * indicates where two variables are statistically significant. Sample sizes vary across comparisons because some participants had missing data for specific variables. Values less than 291 for the no injury group or less than 11 for the serious injury group indicate that data for that variable were not available for all individuals in the respective injury status category. For “falls/year × sex,” Levene’s test indicated unequal variances (F = 7.13, *p* = 0.008), so Welch’s *t*-test was used; all other tests assumed equal variances.

**Table 7 healthcare-13-03226-t007:** Clinical variables and outcomes used and their relationships analysed through one-way ANOVA and post hoc comparisons where necessary.

Dependent Variable	Outcomes/Factors	F(df1, df2)	*p*-Value
Falls/year	Diagnosis (functional, delirium, or dementia)	13.56 (2, 297)	<0.001 *
Mean HGS (both hands)	Destination placement	2.11 (5, 224)	0.060
Tinetti score	Destination placement	7.95 (5, 213)	<0.001 *
MRC-SS	Destination placement	2.92 (5, 207)	0.014 *
Falls/year	Destination placement	6.76 (5, 224)	<0.001 *
Falls/year	Intervention type	5.16 (2, 298)	0.006 *
Length of stay	Intervention type	1.19 (2, 298)	0.306
Falls/year	Ward	5.49 (5, 294)	<0.001 *

Note: * indicates where two variables are statistically significant.

**Table 8 healthcare-13-03226-t008:** Variables compared and chi-squared test results.

Variable Compared	χ^2^	df	*p*-Value
Destination placement × Intervention	20.7	10	0.023 *
Serious injury × Diagnosis	0.77	2	0.680
Falls per year × Sex	119.35	124	0.601

Note: * indicates where two variables are statistically significant.

## Data Availability

The raw data presented in this article will be made available by the authors upon request. The data is not publicly available due to privacy restrictions.

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
