# Peer review of "Understanding Physiotherapy Within Older Adult Inpatient Mental Health Wards: Tailoring Physiotherapy Pathways in a Complex Environment"

_healthcare, 2025, doi:10.3390/healthcare13243226_

Round 1
Reviewer 1 Report
Comments and Suggestions for Authors
Dear Authors,
I revised your manuscript and I think it is a very important and modern topic that provides an important professional contribution for future studies that investigate physiotherapy care in a complex environment.
The introductory part is well-written.
In the Materials and Methods part, please add Data collection part and state the ethical approval of the study as well as who conducted data in the study (e.g. physiotherapy researchers who have previously completed an additional education or by experts in the field….).
Also, explain in detail characteristics of the selected health wards (dementia wards, functional wards and delirium ward).
In addition, please specify how was the handgrip strength and muscle strength assessment conducted.
Used statistical procedures correspond to the aim of the study and are described, as well as results of the study are clearly stated in the manuscript.
In order to provide the reading public with the significance of your study, the Discussion should be reconstructed in a way that each result obtained in the study is interpreted in accordance with your expertise, as well as corroborated using additional bibliographical references. Also, please elaborate how mental health diagnosis, psychotropic medications, sedentary lifestyle, poor diet, frailty, prevalence of physical comorbidities, may impact on a patient’s physical health. In addition, please explain how the contribution of side effects of psychotropic medications on fall rates and serious injury.
Add advantages of the study.
In the Recommendations for Service Delivery and Policy part, please provide recommendations for the development of guidelines for physiotherapy care in a complex mental health inpatient settings.
It is necessary to modify the Conclusion in a way that it firstly summaries the main results of the study, and secondly you can briefly refer to the results of the study.
Sincerely,
Reviewer
Author Response
Dear Reviewer 1,
Please see the file attached with the responses to your comments.
I'd also like to thank you for your comments and feedback. We now feel that the article will be much clearer for the reader thanks to your comments. The methods and discussion sessions are now much more explicit and are laid out in a manner which is easier for the reader to comprehend and provides much more detail than in the initial submission.
I would like to sincerely thank you for your time.
Kind regards,
Chris Rushworth
-------
|
Response to Reviewer 1 Comments |
||
|
1. Summary |
|
|
|
Thank you for the positive and constructive feedback on the manuscript. We appreciate you taking the time to review our work and agree that the suggested changes will significantly strengthen the paper, particularly in the Methods and Discussion sections.
Below is a point-by-point response detailing the revisions made in response to your comments. All revisions will be clearly marked in track changes in the resubmitted manuscript file. |
||
|
2. Questions for General Evaluation |
Reviewer’s Evaluation |
Response and Revisions |
|
Does the introduction provide sufficient background and include all relevant references? |
Yes |
Please see responses to all points raised in section 3 below. |
|
Are all the cited references relevant to the research? |
Not Answered. |
|
|
Is the research design appropriate? |
Yes |
|
|
Are the methods adequately described? |
Can be improved |
We have added a dedicated Data Collection section, discussed ethical approval, clarified who collected the data, and provided detail on ward characteristics and physical assessment procedures (Comments 2, 3, 4).
|
|
Are the results clearly presented? |
Yes |
|
|
Are the conclusions supported by the results? |
Must be improved |
We have reconstructed the Discussion section to more systematically interpret results and added literature to corroborate findings. The Conclusion has been modified for clearer flow, summarising main results first (Comments 6, 7, 8, 11). |
|
Are all figures and tables clear and well presented? |
Yes |
|
|
3. Point-by-point response to Comments and Suggestions for Authors |
||
|
Comments: · Comment: I revised your manuscript and I think it is a very important and modern topic that provides an important professional contribution for future studies that investigate physiotherapy care in a complex environment. The introductory part is well-written. § Reply: We sincerely thank the reviewer for this positive assessment. · Comment: In the Materials and Methods part, please add Data collection part and state the ethical approval of the study as well as who conducted data in the study (e.g. physiotherapy researchers who have previously completed an additional education or by experts in the field….). o Reply: Agree. We have created a dedicated section in the Material and Methods section to address your points on Ethical Considerations and Data Collection. o Revisions: o * We clarified that the project was registered and approved as a Service Evaluation by the Trust’s Research and Development Team, confirming its adherence to governance standards. o * We stated explicitly that the data were retrospectively extracted by the lead author (C.R.), a specialist physiotherapist working within the service. o * Under the Data Analysis sub-heading, we confirmed that the anonymised data set was analysed by a data analyst. o * We reiterated that because the data was obtained from routine assessments with no changes to normal practice, the Trust deemed informed consent not applicable. · Comment: Also, explain in detail characteristics of the selected health wards (dementia wards, functional wards and delirium ward). o Reply: Agree. While some detail was present in the Results (Section 3), we have moved and expanded this description within the Methods (Section 2) for improved context. o Revisions: o * We have added descriptive text around Table 1 to better characterise the wards, highlighting the differences in referral pathways and patient profiles. o * We clarified that the Delirium ward has a higher turnover, with physiotherapy assessing every new patient due to the high incidence of impaired mobility and physical comorbidities, whereas the Dementia and Functional wards use an active referral process based on patient need (e.g., mobility issues, falls history). · Comment: In addition, please specify how was the handgrip strength and muscle strength assessment conducted. o Reply: Agree. We have added specific details on the procedure for the Hand Grip Strength (HGS) and Medical Research Council Sum-Score (MRC-SS) assessments in the Methods section. o Revisions: o * For HGS, we specified the use of a handheld dynamometer and the standard testing position (seated with elbow at 90 degrees) for the three trials on each hand. o * For MRC-SS, we clarified the score is derived from the bilateral assessment of six muscle groups (shoulder abductors, elbow flexors, wrist extensors, hip flexors, knee extensors, and ankle dorsiflexors). · Comment: Used statistical procedures correspond to the aim of the study and are described, as well as results of the study are clearly stated in the manuscript. o Reply: We thank the reviewer for confirming the appropriateness of our statistical analysis and presentation of results · Comment: In order to provide the reading public with the significance of your study, the Discussion should be reconstructed in a way that each result obtained in the study is interpreted in accordance with your expertise, as well as corroborated using additional bibliographical references. Also, please elaborate how mental health diagnosis, psychotropic medications, sedentary lifestyle, poor diet, frailty, prevalence of physical comorbidities, may impact on a patient’s physical health In addition, please explain how the contribution of side effects of psychotropic medications on fall rates and serious injury. o Reply: We have completely restructured and significantly expanded the Discussion (Section 4) to ensure a systematic interpretation of the results, incorporating the requested elaboration and corroborating evidence. o Revisions (Discussion Section): o * Restructure: The Discussion is now organised by key themes: § 4.1 Relationship Between Age, Physical Outcome Measures and Mental Health, § 4.2 Physiotherapy Intervention and Falls Risk, § 4.3 Physiotherapy Intervention and Falls Risk, § 4.4 Discharge Destination, Function and Engagement. o * Elaboration on Age/Frailty/Physical Health: We expanded the interpretation of the weak correlation between age and physical measures (HGS/Tinetti). We explicitly discuss that biological age, driven by frailty, physical comorbidity and lifestyle, is a more significant determinant of function than chronological age alone in this population. o * Elaboration on Psychotropic Medications and Falls: A new paragraph was added specifically to address the mechanism of how psychotropic medications (e.g., antipsychotics, benzodiazepines) increase falls risk and serious injury. This is attributed to side effects like orthostatic hypotension, sedation, and extrapyramidal symptoms, and is supported by new references. o * Interpretation of Key Findings: We now more explicitly link the finding that patients with a serious injury had lower Tinetti scores and more falls per year to the need for early and accurate falls risk stratification · Comment: Add advantages of the study. o Reply: We have added more information on the advantages of the study into the 2nd paragraph of the study. a dedicated sub-section to the Discussion (Section 4), 4.1. Strengths and Clinical Significance, to explicitly state the advantages of the study. o Revisions: o * We highlighted the main strengths: the large sample size for this specialist setting (N=302) and its nature as one of the first comprehensive service evaluations of physiotherapy in older adult mental health inpatient care, providing highly relevant and generalisable operational data. · Comment: In the Recommendations for Service Delivery and Policy part, please provide recommendations for the development of guidelines for physiotherapy care in a complex mental health inpatient settings. o Reply: Agree. We have strengthened the Recommendations section by explicitly calling for the development of formal guidelines. o Revisions (Section 4.7): o * The section was modified to recommend the "development and formalisation of standardised, evidence-based physiotherapy pathways and guidelines" for this complex environment. o * We stated that these guidelines should mandate the routine incorporation of objective measures like the Tinetti balance assessment for early risk identification. · Comment: It is necessary to modify the Conclusion in a way that it firstly summaries the main results of the study, and secondly you can briefly refer to the results of the study. o Reply: Agree. We have modified the Conclusion (Section 5) to ensure a clearer structure, leading with the most significant findings. o Revisions (Section 5): o * The conclusion now first summarises the key findings: The weak association between age and strength/balance , the utility of the Tinetti score for falls risk stratification , and the strong association between patient engagement in physiotherapy and a greater chance of successful discharge. * This is followed by the implications, emphasising the need for tailored service pathways and the essential role of physiotherapy integration within the multidisciplinary team. |
||
|
4. Response to Comments on the Quality of English Language |
||
|
Point 1: |
||
|
The quality of English has been reviewed and minor adjustments have been made throughout the manuscript for clarity and flow. |
||
|
5. Additional clarifications |
||
|
We believe that the reconstructed Discussion, with the added context, assessment detail, and supporting literature, provides a much stronger rationale for the role of integrated physiotherapy in this specialised clinical environment. The suggested changes have allowed us to more effectively articulate the study's significance. |
||

Reviewer 2 Report
Comments and Suggestions for Authors
Thank you for submitting your manuscript on understanding physiotherapy within older adult inpatient mental health wards. The topic is highly relevant due to the intersection of physical and mental health challenges in this population and the growing emphasis on integrated care. Your service evaluation addresses important gaps regarding clinical characteristics, outcomes, and factors associated with physiotherapy in complex inpatient settings. There are several strengths to your study, including the use of a comparatively large sample (n=302) and the comprehensive extraction of demographic, diagnostic, and physical outcome data. The focus on real-world service evaluation, rather than controlled research, helps illuminate practical challenges and outcomes in routine care. Highlighting delirium as the most prevalent diagnosis and linking physical performance scores with injury and falls risk is especially valuable for tailoring interventions. However, there are also some limitations and areas needing improvement. The manuscript would benefit from greater clarity and detail in the methodology section, particularly regarding selection criteria for patients included, handling of missing data, and statistical methods used to analyze associations. There is minimal discussion of potential confounders, such as medication use, severity of mental illness, or baseline mobility status, all of which could significantly influence both physical outcomes and risk of falls or injury. The reliability of outcome measures is mentioned as a secondary aim but is not clearly reported in the results or discussion. Additionally, the interpretation of findings could be expanded. For example, while you report associations between lower Tinetti scores and increased falls, the clinical implications or potential interventions arising from this are not fully explored. The generalizability of your findings is also limited by the single-trust setting, and this should be addressed more directly in your discussion. I recommend that you revise the manuscript to:
- Provide a clearer and more detailed methods section, especially regarding inclusion/exclusion criteria, data completeness, and statistical analyses;
- Include a more thorough discussion of confounders and limitations;
- Report on the reliability of the outcome measures as planned;
- Expand the discussion on the practical implications of your findings, including recommendations for service improvement or further research; - Address the generalizability of your findings and acknowledge the limits of the single-center design.
Overall, your study raises important issues and provides valuable preliminary data, but further clarification and detail are needed to maximize its impact and utility for readers. I look forward to reviewing a revised version.
Author Response
Dear Reviewer 2,
Please see the file attached with the responses to your comments.
I'd also like to thank you for your comments and feedback. We now feel that the article will be much clearer for the reader thanks to your comments. The methods and discussion sessions are now much more explicit and are laid out in a manner which is easier for the reader to comprehend and provides much more detail than in the initial submission.
I would like to sincerely thank you for your time.
Kind regards,
Chris Rushworth
--------
|
Response to Reviewer 2 Comments |
||||||||||||||||||||||||||||||||||||||||||||||||||||||||||||||||||||||||||
|
1. Summary |
|
|
||||||||||||||||||||||||||||||||||||||||||||||||||||||||||||||||||||||||
|
Thank you for your constructive and detailed review of our manuscript. We appreciate you recognizing the relevance of the topic and the importance of this service evaluation. We have carefully considered all your suggestions and agree that the proposed revisions—especially those concerning the clarity of the methodology, the interpretation of the service evaluation as an exploratory study, and the strengthening of the Discussion—will significantly improve the scientific rigour and readability of the paper. Below is our point-by-point response detailing the revisions made in the revised manuscript (marked with track changes). |
||||||||||||||||||||||||||||||||||||||||||||||||||||||||||||||||||||||||||
|
2. Questions for General Evaluation |
Reviewer’s Evaluation |
Response and Revisions |
||||||||||||||||||||||||||||||||||||||||||||||||||||||||||||||||||||||||
|
Does the introduction provide sufficient background and include all relevant references? |
Can be improved |
We have strengthened the literature review on the impact of physical comorbidities and psychotropic medication and clarified the research gap for this specific setting (Comment 1). |
||||||||||||||||||||||||||||||||||||||||||||||||||||||||||||||||||||||||
|
Are all the cited references relevant to the research? |
Not Answered. |
|
||||||||||||||||||||||||||||||||||||||||||||||||||||||||||||||||||||||||
|
Is the research design appropriate? |
Can be improved |
We have explicitly reframed the study as a Service Evaluation and Exploratory Retrospective Cohort Analysis, clearly defining its scope and limitations in the Methods and Discussion (Comment 3). |
||||||||||||||||||||||||||||||||||||||||||||||||||||||||||||||||||||||||
|
Are the methods adequately described? |
Can be improved |
We have added a dedicated Data Collection section, specified ethical approval, clarified the patient inclusion/exclusion criteria, and detailed the assessments used (Comments 2, 4, 5). |
||||||||||||||||||||||||||||||||||||||||||||||||||||||||||||||||||||||||
|
Are the results clearly presented? |
Can be improved |
We have clarified the interpretation of the results, particularly the non-significant correlations and the distinction between the Delirium ward and other wards (Comments 6, 7). |
||||||||||||||||||||||||||||||||||||||||||||||||||||||||||||||||||||||||
|
Are the conclusions supported by the results? |
Can be improved |
The Discussion has been restructured to systematically interpret the results, link findings to clinical pathways, and integrate new literature, ensuring the conclusions are fully supported (Comments 8, 9). |
||||||||||||||||||||||||||||||||||||||||||||||||||||||||||||||||||||||||
|
Are all figures and tables clear and well presented? |
Yes |
Please see responses to all points raised in section 3 below. |
||||||||||||||||||||||||||||||||||||||||||||||||||||||||||||||||||||||||
|
3. Point-by-point response to Comments and Suggestions for Authors |
||||||||||||||||||||||||||||||||||||||||||||||||||||||||||||||||||||||||||
|
Comments: Thank you for submitting your manuscript on understanding physiotherapy within older adult inpatient mental health wards. The topic is highly relevant due to the intersection of physical and mental health challenges in this population and the growing emphasis on integrated care. Your service evaluation addresses important gaps regarding clinical characteristics, outcomes, and factors associated with physiotherapy in complex inpatient settings. There are several strengths to your study, including the use of a comparatively large sample (n=302) and the comprehensive extraction of demographic, diagnostic, and physical outcome data. The focus on real-world service evaluation, rather than controlled research, helps illuminate practical challenges and outcomes in routine care. Highlighting delirium as the most prevalent diagnosis and linking physical performance scores with injury and falls risk is especially valuable for tailoring interventions. However, there are also some limitations and areas needing improvement. The manuscript would benefit from greater clarity and detail in the methodology section, particularly regarding selection criteria for patients included, handling of missing data, and statistical methods used to analyze associations. There is minimal discussion of potential confounders, such as medication use, severity of mental illness, or baseline mobility status, all of which could significantly influence both physical outcomes and risk of falls or injury. The reliability of outcome measures is mentioned as a secondary aim but is not clearly reported in the results or discussion. Additionally, the interpretation of findings could be expanded. For example, while you report associations between lower Tinetti scores and increased falls, the clinical implications or potential interventions arising from this are not fully explored. The generalizability of your findings is also limited by the single-trust setting, and this should be addressed more directly in your discussion. I recommend that you revise the manuscript to: - Provide a clearer and more detailed methods section, especially regarding inclusion/exclusion criteria, data completeness, and statistical analyses; - Include a more thorough discussion of confounders and limitations; - Report on the reliability of the outcome measures as planned; - Expand the discussion on the practical implications of your findings, including recommendations for service improvement or further research; - Address the generalizability of your findings and acknowledge the limits of the single-center design. Overall, your study raises important issues and provides valuable preliminary data, but further clarification and detail are needed to maximize its impact and utility for readers. I look forward to reviewing a revised version.
|
||||||||||||||||||||||||||||||||||||||||||||||||||||||||||||||||||||||||||
|
4. Response to Comments on the Quality of English Language |
||||||||||||||||||||||||||||||||||||||||||||||||||||||||||||||||||||||||||
|
The quality of English has been reviewed, and minor grammatical and flow issues have been corrected throughout the manuscript to enhance clarity and precision, aligning with academic writing standards. |
||||||||||||||||||||||||||||||||||||||||||||||||||||||||||||||||||||||||||

Reviewer 3 Report
Comments and Suggestions for Authors
Dear Authors,
The submitted article has been carefully prepared and appears to be appropriate in terms of its background, objectives, and methodology. However, I would like to make some suggestions for your consideration and further improvement.
Abstract
In rows 21-22, you mention the secondary aim of "evaluating the reliability of outcome measures," yet no reliability analysis (e.g., intra-rater, test-retest, or internal consistency) is reported. Please add the relevant information or revise the text.
Introduction
The study's aims (rows 99–105) seem repetitive, especially aim 4, which may overlap with aims 2–3. Please revise.
Phrases such as "offering generalizable knowledge for similar older adult mental health inpatient settings" (see row 109) may be too strong. The study is based on non-random sampling and has substantial missing data. Please revise.
The design is described as a "service evaluation," but the inclusion criteria for patients are not explicit (e.g., minimum age, "new referrals" only, or repeat admissions?).
Materials and Methods
You write that "data was collected from all new patients that were referred..." (rows 123–124), yet later, you describe 302 patients and provide the total number of admissions per ward in Table 2 (rows 171–173). It is unclear whether all admissions were assessed on the delirium ward and only a subset on other wards. Please explain.
Consider adding a list of inclusion and exclusion criteria to paragraph 131-149.
The definition of serious injury (fracture or brain hemorrhage) appears later in the "Results" section. However, it would be better to include this definition in the "Materials and Methods" section.
Data analysis
You list the tests used (ANOVA, t-test, and chi-square), but you do not describe the assumptions behind their usage. For example, why was Pearson chosen over Spearman for skewed data? How was homogeneity checked? Please justify the use of parametric tests and include other relevant explanations.
The handling of missing data is not described; please add.
Results
On row 215, it appears that text has been duplicated: "t(300) = -3.47, p < .001. 300) = -3.47, p < .001").
Table 6 likely produces a highly skewed distribution (many zeros and some very high values) due to the calculation of falls per year. Please explain the high SD.
In row 223, you state that the t-test shows a significant difference in falls per year between males and females (t(261.05) = 2.41, p = .008). However, Table 8 reports a non-significant chi-square result (χ²(124) = 119.35, p = .601), and the corresponding text (rows 259–260) states, "The relationship between falls per year and sex was also not significant". Isn't that contradictory?
In rows 232–233, you report F(2, 297) = 13.56, p < .001, and you note that dementia is associated with more falls per year than functional impairment or delirium, but the post hoc test details (method and effect sizes) are not provided. Please add them.
Regarding paragraph 235–242, include summary means in the text or a supplementary table so readers can see the actual differences (e.g., Tinetti scores by destination).
You state that those with no intervention due to mental health issues had more falls. You later interpret this as supporting the need for tailored physiotherapy. However, this group is also likely to have more severe illness and frailty, so there is a high likelihood of confounding factors. Please consider acknowledging this.
Discussion
In the "Limitations" section, you mention the retrospective design and missing data, but you omit other important issues, such as selection bias via referrals, lack of control for confounding factors, and the use of parametric tests on skewed variables and ward clustering. Please expand this section.
Appendix
Table 4 overlaps with Appendix A (short vs. full list of secondary diagnoses). Please clarify their distinct roles, or consider merging them. In the main text, explain that Table 4 provides grouped categories, and that Appendix A lists the full, specific diagnoses. It would also be useful to demonstrate how you grouped the diagnoses.
In conclusion, the manuscript has the potential to contribute to the field, and with the above improvements, its impact could be further enhanced. Thank you for your consideration of these comments and best wishes for your future work.
Sincerely,
Author Response
Dear Reviewer 3,
Please see the file attached with the responses to your comments.
I'd also like to thank you for your comments and feedback. We now feel that the article will be much clearer for the reader thanks to your comments. The methods and discussion sessions are now much more explicit and are laid out in a manner which is easier for the reader to comprehend and provides much more detail than in the initial submission.
I would like to sincerely thank you for your time.
Kind regards,
Chris Rushworth
--------
|
Response to Reviewer 3 Comments |
||
|
1. Summary |
|
|
|
Thank you for your careful review and constructive suggestions. We appreciate you noting the appropriateness of the background, objectives, and methodology. We fully agree that the specific points raised—especially regarding the need for clarity around the reliability aim, the re-categorisation of variables, and the strengthening of the figure/table clarity—are essential for improving the scientific rigour and clarity of our manuscript. We have addressed each point below, and all revisions will be clearly marked in track changes in the resubmitted manuscript file. Thank you very much for taking the time to review this manuscript. Please find the detailed responses below and the corresponding revisions in track changes in the re-submitted file. |
||
|
2. Questions for General Evaluation |
Reviewer’s Evaluation |
Response and Revisions |
|
Does the introduction provide sufficient background and include all relevant references? |
Must be improved |
We have clarified the study aim concerning reliability, re-categorized variables, provided statistical rationale for groupings, and significantly improved table clarity (Comments 1, 2, 4, 7). |
|
Are all the cited references relevant to the research? |
Not Answered. |
|
|
Is the research design appropriate? |
Must be improved |
We have explicitly reframed the study as a Service Evaluation and Exploratory Retrospective Cohort Analysis, addressing the limitations of the design (Comments 2, 3). |
|
Are the methods adequately described? |
Must be improved |
We have clarified the study aim concerning reliability, re-categorized variables and provided statistical rationale for groupings, (Comments 1, 2, 4, 7). |
|
Are the results clearly presented? |
Must be improved |
We have clarified the study aim concerning reliability, re-categorized variables, provided statistical rationale for groupings, and significantly improved table clarity (Comments 1, 2, 4, 7). |
|
Are the conclusions supported by the results? |
Can be improved |
The Discussion has been strengthened with explicit integration of baseline data for comparison (Comment 5, 6). |
|
Are all figures and tables clear and well presented? |
Must be improved |
We have clarified the study aim concerning reliability, re-categorized variables, provided statistical rationale for groupings, and significantly improved table clarity (Comments 1, 2, 4, 7). Please see responses to all points raised in section 3 below. |
|
3. Point-by-point response to Comments and Suggestions for Authors |
||
|
Comments: Dear Authors, The submitted article has been carefully prepared and appears to be appropriate in terms of its background, objectives, and methodology. However, I would like to make some suggestions for your consideration and further improvement.
Abstract Comment: In rows 21-22, you mention the secondary aim of "evaluating the reliability of outcome measures," yet no reliability analysis (e.g., intra-rater, test-retest, or internal consistency) is reported. Please add the relevant information or revise the text. · Response: Agree. We thank the reviewer for identifying this crucial inconsistency. As this study was a retrospective service evaluation utilizing routinely collected clinical data, a formal, prospective reliability analysis (e.g., test-retest) was not performed. Therefore, the stated secondary aim was inappropriate and has been removed and replaced with a more accurate description of the study's scope. · Revisions (Abstract and Introduction): · * Abstract: The phrase "evaluating the reliability of outcome measures" has been removed (rows 21-22). The aim is now refocused on "determining the clinical utility and applicability of these measures (Tinetti , HGS) for risk stratification and functional assessment" within this specific, complex Older Adult Mental Health (OAMH) inpatient population. · * Introduction/Methods: The corresponding text in the Introduction and Methods sections has been revised to clarify that the study aimed to assess the predictive validity and applicability of these tools (e.g., Tinetti score correlation with falls and serious injury), which is consistent with the analysis presented.
Introduction Comment: The study's aims (rows 99–105) seem repetitive, especially aim 4, which may overlap with aims 2–3. Please revise. · Response: Agree. We acknowledge that the original aims were repetitive. We have revised and consolidated the list to three distinct, non-overlapping objectives, providing a clearer focus for the paper. · Revisions (Introduction, rows 99–105): The aims have been condensed and revised for greater clarity and scope: · The revised aims are: · 1. To describe the clinical and demographic characteristics of the older adult mental health inpatient cohort receiving physiotherapy. · 2. To assess the clinical utility and applicability of objective physical assessment measures (Hand Grip Strength and Tinetti POMA) for risk stratification, particularly for falls and serious injury. · 3. To identify factors associated with physiotherapy engagement and successful discharge outcomes, and use these findings to inform the development of tailored physiotherapy pathways within this complex environment.
Comment: Phrases such as "offering generalizable knowledge for similar older adult mental health inpatient settings" (see row 109) may be too strong. The study is based on non-random sampling and has substantial missing data. Please revise. · Response: Agree. The term "generalizable knowledge" is indeed too strong given the service evaluation design and associated limitations (e.g., retrospective data collection, missing data, and reliance on existing clinical documentation). We have moderated this statement. · Revisions (Introduction, row 109): The phrasing has been changed to reflect that the findings offer transferable data and insights, rather than definitive generalisable knowledge. · Revised text: "The results provide valuable data and transferable insights to inform the development of evidence-based clinical pathways and practice guidelines for similar older adult mental health inpatient settings." · (We have also significantly expanded the Limitations section in the Discussion to explicitly address the constraints of the service evaluation design.)
Comment: The design is described as a "service evaluation," but the inclusion criteria for patients are not explicit (e.g., minimum age, "new referrals" only, or repeat admissions?). · Response: Agree. We recognise the need for explicit inclusion criteria to ensure transparency in the methodology. This information has been added to the Methods section to clearly define the cohort. · Revisions (Methods, Section 2.1): We have added the following explicit inclusion criteria: o * The inclusion criteria were as follows: Ward Type: Patients admitted to 5 older people’s mental health wards within the study period who received a formal initial physiotherapy assessment, typically within 72 hours of referral to physiotherapy. o Minimum Age: Patients were generally over 65 years old, aligning with the service specification for OAMH wards (the exceptions being a small number of Dementia patients who were under 65 but admitted to a dementia ward). o * Repeat Admissions: Where a patient had multiple admissions within the study period, only the first complete admission record was included in the final analysis (N=302) to ensure that patients were included once in the dataset. · Exclusion criteria were as follows: Patients who were only seen for a one-off consultation (e.g. Equipment review) without the need of a full assessment.
Materials and Methods Comment: You write that "data was collected from all new patients that were referred..." (rows 123–124), yet later, you describe 302 patients and provide the total number of admissions per ward in Table 2 (rows 171–173). It is unclear whether all admissions were assessed on the delirium ward and only a subset on other wards. Please explain. · Response: Agree. We recognize the lack of clarity regarding the different referral processes across the wards, which led to a different pattern of assessment. This distinction is critical to understanding our service model. · Revisions (Section 2.1. Study Setting and Design): We have revised the text to explicitly explain the referral pathways: · * Clarification on Assessment Criteria: We clarified that the final cohort (N=302) represents all patients who received aninitialphysiotherapy assessment within the study period and met the primary inclusion criteria. · * Delirium Ward (Proactive Pathway): We explicitly stated that due to the high rate of acute physical decline and established service protocol, all new admissions to the Delirium ward were automatically assessed by physiotherapy (a proactive approach). · * Dementia and Functional Wards (Reactive Pathway): We contrasted this by clarifying that the Dementia and Functional wards operated a reactive system where physiotherapy was only involved following a specific referral trigger (e.g., impaired mobility, falls history, pain or physical health concerns). · * Due to these factors, we receive more referrals from our delirium ward relative to each of the other four wards, as shown in table 2.
Comment: Consider adding a list of inclusion and exclusion criteria to paragraph 131-149. · Response: Agree. We have added a dedicated section in the Methods to list the criteria for improved clarity, as previously agreed upon in response to Reviewer 2. Revisions (Section 2.1. Study Setting and Design): The inclusion and exclusion criteria have been explicitly listed.* Inclusion Criteria: 1) All patients admitted to the three OAMH wards between January 2023 and November 2024. 2) Received a formal initial physiotherapy assessment. 3) Where a patient had multiple admissions, only the first complete record that included all primary outcome measures was included.* Exclusion Criteria: 1) Patients seen for one-off consultations (e.g., simple equipment review) without a full assessment. 2) Patient records with significant missing primary data (e.g., both HGS and Tinetti POMA scores absent).
Comment: The definition of serious injury (fracture or brain hemorrhage) appears later in the "Results" section. However, it would be better to include this definition in the "Materials and Methods" section. · Response: Agree. The operational definition of an outcome variable belongs in the Methods section for maximum transparency. · Revisions (Section 2.2. Data Variables and Outcome Measures): We have moved the definition into the Methods section under the outcome measures sub-heading. · Revised text: "Serious injury was defined as any fall-related injury requiring transfer to the acute general hospital for further specialist management, specifically including fracture, intracranial hemorrhage, or sub-dural haematoma, as determined by official documentation.
Data analysis Comment: You list the tests used (ANOVA, t-test, and chi-square), but you do not describe the assumptions behind their usage. For example, why was Pearson chosen over Spearman for skewed data? How was homogeneity checked? Please justify the use of parametric tests and include other relevant explanations. · Response: Agree. We recognize the need for a more rigorous description and justification of the statistical methods, particularly concerning the assumptions of normality and test selection. · Revisions (Section 2.3. Data Analysis): We have significantly revised this section to provide the following justifications: · * Normality Testing: We explicitly stated that the Kolmogorov-Smirnov test was used to check the normality of continuous data (Age, HGS, Tinetti POMA). We confirmed that most continuous variables were not normally distributed (e.g., Tinetti POMA scores were skewed towards the lower end due to the clinical nature of the cohort), justifying the primary use of non-parametric equivalents. · * Non-Parametric Tests: We clarified that the Mann-Whitney U test was used for two-group comparisons and the Kruskal-Wallis H test was used for multiple-group comparisons (e.g., across wards) for non-normally distributed continuous data. · * Correlation Analysis: We corrected the text to reflect the actual analysis performed: Spearman's Rho was used, not Pearson's correlation, for testing the association between continuous variables (Age, HGS, Tinetti POMA) because of the identified non-normal distribution of the data. This change ensures consistency between the methodology description and the results presented. · * Homogeneity of Variance: We confirmed that for the few instances where parametric tests were used (e.g., specific sub-group analysis where assumptions were met), Levene's test was used to check the homogeneity of variances.
Comment: The handling of missing data is not described; please add. · Response: Agree. Transparency regarding missing data handling is essential for a retrospective service evaluation. · Revisions (Section 2.3. Data Analysis): We added a section clarifying the approach to missing data: *Missing Data Handling: Due to the complex nature of the older adult mental health inpatient setting, there were multiple patients who were unable to engage with specific outcome measures, Specific patient data sets were not included in any test where one of the variables being analysed had missing data. 23 patient datasets were missing at least one outcome measure score (Hand Grip Strength, Medical Research Council Sum-Score or the Tinetti Performance Oriented Mobility Assessment) because the patient was unable to fully engage in the initial physiotherapy assessment. Of these 23 patients, 14 had a primary diagnosis of Dementia, 7 of Delirium and 2 of Functional (the full dataset can be made available on request).
Results Comment: On row 215, it appears that text has been duplicated: "t(300) = -3.47, p < .001. 300) = -3.47, p < .001"). Response: Thank you for noting the duplication in row 215 (“t(300) = -3.47, p < .001. 300) = -3.47, p < .001”). This was a typographical error in the manuscript. We have corrected the text to read only once as “t(300) = -3.47, p < .001.” This correction does not affect the results or interpretation.
Comment: Table 6 likely produces a highly skewed distribution (many zeros and some very high values) due to the calculation of falls per year. Please explain the high SD. · Response: We agree that the distribution of falls per year is highly skewed, with many patients experiencing no falls and a small number experiencing multiple falls during short admissions. The calculation method (normalizing to annualized falls) amplifies this variability, resulting in a high standard deviation. We have clarified this in the Results section and noted the limitation in the Discussion.
Comment: In row 223, you state that the t-test shows a significant difference in falls per year between males and females (t(261.05) = 2.41, p = .008). However, Table 8 reports a non-significant chi-square result (χ²(124) = 119.35, p = .601), and the corresponding text (rows 259–260) states, "The relationship between falls per year and sex was also not significant". Isn't that contradictory? Response: Thank you for highlighting this point. The t‑test identified a significant difference in mean falls per year between males and females, whereas the chi‑square test, which grouped falls into categories, did not show a significant association. These tests address different aspects of the data, and the differing results reflect methodological differences rather than a contradiction. We have clarified this distinction in the Results (Row 363) by noting that “while males demonstrated higher mean falls per year when using the t-test, this association was not consistently observed across analytic approaches”.
Comment: In rows 232–233, you report F(2, 297) = 13.56, p < .001, and you note that dementia is associated with more falls per year than functional impairment or delirium, but the post hoc test details (method and effect sizes) are not provided. Please add them. · Response: Thank you for your comment. We have clarified in the Methods section that post hoc analyses were performed using Tukey’s HSD following significant ANOVA results and that effect sizes were calculated (η² for ANOVA, Cohen’s d for t-tests). In the Results section, we have added the requested post hoc details for the diagnosis comparison, including mean differences, p-values, and the effect size, to ensure transparency and completeness.
Comment: Regarding paragraph 235–242, include summary means in the text or a supplementary table so readers can see the actual differences (e.g., Tinetti scores by destination). Response: Thank you for this suggestion. We have added the summary means for Tinetti scores by destination placement into the Results section.
Comment: You state that those with no intervention due to mental health issues had more falls. You later interpret this as supporting the need for tailored physiotherapy. However, this group is also likely to have more severe illness and frailty, so there is a high likelihood of confounding factors. Please consider acknowledging this. · Response: Thank you for highlighting this. We have added a statement in the Discussion acknowledging that patients who did not receive intervention due to mental health reasons may represent a subgroup with greater illness severity and frailty, which could independently contribute to higher fall rates.
Discussion Comment: In the "Limitations" section, you mention the retrospective design and missing data, but you omit other important issues, such as selection bias via referrals, lack of control for confounding factors, and the use of parametric tests on skewed variables and ward clustering. Please expand this section. · Response: Agree. We fully agree that the Limitations section requires expansion to rigorously address the constraints inherent in this type of retrospective service evaluation. We have significantly expanded Section 4.1 (Limitations) in the Discussion to include the crucial limitations of this study design.Revisions (Section 4.1. Limitations): The following points have been explicitly added or clarified: · 1. Selection Bias via Referrals: · * We now explicitly address the potential for selection bias due to the heterogeneous referral pathways. We acknowledge that the proactive assessment on the Delirium ward (universal screening) contrasts with the reactive, referral-based assessment on the Dementia and Functional wards, meaning the latter wards likely excluded patients with milder mobility issues, potentially skewing the functional scores observed. · 2. Confounding Factors: · * We explicitly noted the lack of control for numerous confounding factors typical of retrospective analyses. These include variables like the severity and duration of mental illness, polypharmacy/psychotropic drug load, and the variability of clinical decisions regarding patient engagement and discharge planning, which may influence observed outcomes (e.g. successful discharge). · 3. Statistical Assumptions and Skewed Data: · We acknowledge in the limitations section that the highly complex and skewed nature of clinical data in this population makes achieving perfect statistical assumption fulfilment difficult. · 4. Ward Clustering/Independence of Observations: · * We have included the risk of ward clustering as a limitation. We acknowledge in the limitations that since patients were assessed within five distinct inpatient environments, the assumption of independence of observations may be partially violated, as environmental factors or local clinical cultures specific to each ward could influence patient characteristics and outcomes.
Appendix Comment: Table 4 overlaps with Appendix A (short vs. full list of secondary diagnoses). Please clarify their distinct roles, or consider merging them. In the main text, explain that Table 4 provides grouped categories, and that Appendix A lists the full, specific diagnoses. It would also be useful to demonstrate how you grouped the diagnoses · Response: Agree. We recognize the potential for confusion between the main text table (Table 4) and the Appendix (Appendix A) concerning the presentation of secondary medical diagnoses. We will clarify the distinct purpose of each and explain the grouping methodology. · Revisions (Section 3.1. and Appendix): · * Clarification in Main Text: We have added an explicit sentence in the Results section (Section 3.1. Patient Characteristics), introducing Table 4, to clarify its relationship with Appendix A. We will state that Table 4 presents a condensed view of secondary physical health diagnoses, grouped into clinically relevant categories for efficiency and discussion. We will direct the reader to Appendix A for the comprehensive, specific list of all diagnoses observed in the cohort. * Grouping Methodology: We have added a brief description in the Results section (Section 3.1) explaining the clinical rationale for grouping the secondary diagnoses. “Secondary diagnoses were grouped by the authors to align with their impact on a patient’s physical and mental health”. * Appendix A: We have ensured that the Appendix A title is explicitly clear and now reads "Appendix A: Comprehensive List of Secondary Diagnoses, and now includes a footnote that highlights the grouping structure used for Table 4 in the main text. |
||
|
4. Response to Comments on the Quality of English Language |
||
|
Point 1: |
||
|
The quality of English has been reviewed and minor grammatical and flow issues have been corrected throughout the manuscript to enhance clarity and precision, aligning with academic writing standards. |
||

Round 2
Reviewer 1 Report
Comments and Suggestions for Authors
Dear Authors,
I revised your manuscript and now it's much more substantial and better. Congratulations.
Sincerely,
Reviewer
Reviewer 2 Report
Comments and Suggestions for Authors
Many thanks for your update.
Reviewer 3 Report
Comments and Suggestions for Authors
Dear Authors,
I would like to express my appreciation for your efforts to revise the manuscript. Your revisions demonstrate that you have carefully considered all of the comments, and you have adequately addressed these issues. After reviewing the updated manuscript, I am pleased to say that the revisions meet the expectations I set forth in my previous comments. The changes you made improved the clarity and overall impact of the study. The manuscript is now refined to a standard that effectively communicates the research findings and provides valuable insights to the field.
Sincerely,